# Ultrasonographic Presentation and Anatomic Distribution of Lung Involvement in Patients with Rheumatoid Arthritis

**DOI:** 10.3390/diagnostics13182986

**Published:** 2023-09-18

**Authors:** Marie Vermant, Alexandros Kalkanis, Tinne Goos, Heleen Cypers, Nico De Crem, Barbara Neerinckx, Veerle Taelman, Patrick Verschueren, Wim A. Wuyts

**Affiliations:** 1Laboratory of Respiratory Diseases and Thoracic Surgery (BREATHE), Department of Chronic Diseases and Metabolism, Katholieke Universiteit Leuven, Herestraat 49, 3000 Leuven, Belgiumwim.wuyts@uzleuven.be (W.A.W.); 2Department of Pulmonology, University Hospitals Leuven, 3000 Leuven, Belgium; 3Department of Rheumatology, University Hospitals Leuven, 3000 Leuven, Belgium; 4Skeletal Biology and Engineering Research Center, Department of Development and Regeneration, KU Leuven, 3000 Leuven, Belgium

**Keywords:** extra-articular manifestations of rheumatoid arthritis, lung ultrasound, rheumatoid-arthritis-associated interstitial lung disease

## Abstract

Background: Rheumatoid arthritis (RA) is a chronic auto-immune disease, typically affecting the joints, which can also present with lung involvement (pleuritis, interstitial lung disease, pulmonary nodules, etc.). Lung ultrasound (LUS) is an upcoming tool in the detection of these pulmonary manifestations. Methods: We performed a 72-window LUS in 75 patients presenting to the outpatient rheumatology clinic and describe the abnormalities (presence of B-lines (vertical comet-tail artefacts), pleural abnormalities, pleural effusions, and subpleural nodules) on lung ultrasound. We created a topological mapping of the number of B-lines per intercostal zone. Results: We observed pleural effusions, pleural abnormalities, and pleural nodules in, respectively, 1.3%, 45.3%, and 14% of patients. There were 35 (46.7%) patients who had less than 5 B-lines, 15 (20%) patients who had between 5 and 10 B-lines, 11 (14.6%) between 10 and 20, 10 (13.3%) between 20 and 50, 1 (1.3%) between 50 and 100, and 3 (4%) of patients who had more than 100 B-lines. Conclusions: LUS in patients with RA shows an array of abnormalities ranging from interstitial syndromes to pleural abnormalities, subpleural nodules, and pleural effusions. Hotspots for the presence of B-lines are situated bilaterally in the posterior subscapular regions, as well as the anterior right mid-clavicular region.

## 1. Introduction

Rheumatoid arthritis (RA) is a chronic autoimmune disease primarily affecting the joints. However, it is increasingly recognized that RA can also have various extra-articular manifestations, including in the lung. Pulmonary involvement in RA can range from pleuritis and interstitial lung disease (ILD) to pulmonary nodules and vasculitis [1]. Traditionally, imaging modalities such as chest X-rays and high-resolution computed tomography (HRCT) scans have been used to rule out or diagnose pulmonary manifestations [1]. Extra-articular manifestations of RA affecting the lung typically follow the development of articular disease, but in rare cases, lung involvement presents as the first evidence of RA and is the predominant feature of the disease [2].

Rheumatoid-arthritis-associated interstitial lung disease (RA-ILD) is estimated to affect approximately 10% of rheumatoid arthritis patients, although there are only a few studies investigating this in detail [3]. Radiographic abnormalities are present in 19% to 57% of RA patients on high-resolution CT, and these could be suggestive of preclinical ILD, in the absence of symptoms [4]. In contrast to the other connective-tissue-disease-related (CTD)-ILDs and similar to idiopathic pulmonary fibrosis, the predominant pattern in RA-ILD is usual interstitial pneumonia (UIP). This pattern is characterized by honeycombing, reticular opacities, and traction bronchiectasis [5]. A UIP pattern is associated with a poor prognosis and outcome, similar to idiopathic pulmonary fibrosis (IPF), resulting in a median survival after diagnosis of only 3–10 years [4,6]. Other radiographic patterns such as non-specific interstitial pneumonia (NSIP) and organizing pneumonia (OP) are associated with a better prognosis than a UIP pattern. Despite improvement in disease-modifying anti-rheumatic drug (DMARD) RA treatments, RA-ILD-associated mortality remains high [7]. Johnson et al. (2023) reported that, even though the overall mortality gap in rheumatoid arthritis is closing, it persists in patients with RA-ILD [8]. As RA-ILD is currently diagnosed in a late stage or during an acute exacerbation, an effective screening program is primordial to detect early disease stages and could significantly impact survival and quality of life, as recent advances have also proven that a diagnostic delay is associated with worse survival [9].

Pulmonary nodules [1] are typically asymptomatic, but can be complicated by cavitation or rupture, leading to infections, pleural effusions, or bronchopleural fistulas. On HRCT, approximately 1% of patients have pulmonary nodules. In contrast, on autopsy, the prevalence rises to 30%. Cavitation of these nodules can be associated with the disease itself or with DMARD therapy [10].

Bronchiectasis is detected by HRCT in approximately 30% of all RA patients and is associated with chronic and recurrent infections, often leading to a vicious circle of disease exacerbations of RA, leading to treatment escalation associated with increased infection risk [1,10,11]. Bronchiolitis, small airway involvement, is more heavily debated, as this disease entity is heavily influenced by smoking history and the presence of RA-ILD. In non-smoking patients, its prevalence is estimated to be 8% based on pulmonary function testing and up to 30% based on HRCT [10,12,13].

Vasculitis is an extremely rare, yet severe complication of rheumatoid arthritis, which typically affects seropositive (anti-citrullinated-protein-antibody (ACPA) and rheumatoid-factor (RF)-positive) patients with long-standing, severe, erosive disease [14,15]. Due to the advances in anti-rheumatic therapies, RA vasculitis has become even less prevalent over the last few decades [16,17].

In addition to the parenchyma, the airways, andathe lung vasculature, the pleura can also be affected by rheumatic disease [18]. Pleural effusions are typically exudates with a secondary evolution towards a fibrothorax. They typically affect rheumatoid-factor-positive, male patients. The prevalence of pleural involvement in RA is between 3 and 5%, and it is present in up to 70% of patients in autopsy studies [1].

Lung ultrasound (LUS) is a low-cost imaging modality, which avoids radiation. Various LUS qualitative and semi-quantitative scores have been studied for the evaluation of lung abnormalities in patients with CTDs and have proven the value of LUS in the detection of pleural and lung abnormalities. In the presence of an interstitial syndrome, such as an interstitial lung disease (ILD), LUS shows the presence of B-lines [19,20,21,22,23,24]. B-lines are vertical artifacts arising from the pleural line, deepening posteriorly and moving synchronously with respiration [25]. LUS has demonstrated a high sensitivity in the detection of signs indicative of ILD, even in the early stages of CTDs and, especially, a high negative predictive value [26]. Furthermore, LUS can be an important tool in the detection of pleural effusions and pleural abnormalities and nodules [27,28].

In this study, we aimed to assess the feasibility of LUS in an outpatient rheumatology clinic in an asymptomatic population of patients with RA and to describe the abnormalities we found upon sonographic evaluation.

## 2. Methods

There were 75 consecutive patients with RA evaluated in the outpatient Rheumatology Clinic of the University of Leuven from January 2023 to June 2023. Inclusion criteria were a diagnosis of RA, confirmed by an expert rheumatologist, according to the 2010 ACR/EULAR classification criteria [29]. No screening for pulmonary symptoms was performed before inclusion.

This study presents the first preliminary results of a larger cross-sectional trial, which has been approved by the Ethical Committee of University Hospitals Leuven (S66850).

### 2.1. Lung Ultrasound 

Lung ultrasound was performed using a Philips Lumify portable ultrasound system and a curved 3.5 MHz array probe by an experienced examiner. To correctly identify the artifactual images of the lungs, the harmonic imaging was removed, and the reject post-processing was lowered. The focus was set at the level of the pleural line, and the depth was set at 12 cm from the pleural line.

Thoracic ultrasound was performed by one sonographer (M.V.), who received training from the European Respiratory Society. The technical aspects of the examination and the quality of the studies were evaluated and controlled by an expert sonographer (A.K.).

The ultrasonographic examinations were performed with patients in the sitting position by moving the probe longitudinally along anatomical reference lines following the well-defined method, published by Gargani et al. and Hassan et al. [30,31]. A novel Android application was designed by the investigators to act as an anatomic map and a guiding system and to improve the standardization of the procedure and the calculation of the LUS score. The comprehensive US assessment was performed in a total of 72 lung intercostal spaces (LISs).

The assessment of the anterior right chest was performed from the second to fifth LIS along the para-sternal, mid-clavicular, axillary anterior, and mid-axillary chest lines, whereas an assessment from the second to fourth LIS along the same lines was performed for the anterior left chest. An assessment of the left fifth LIS was not performed, since the heart blocks the correct visibility of the wall interface.

At the posterior chest level, the US examination was obtained from the second to tenth LIS along the paravertebral lines and from the seventh to eighth LIS along both the sub-scapular and axillary posterior lines. Each LIS was scanned in a longitudinal scan moving the probe from the medial to lateral part along the anatomical references lines to enable maximum coverage of the anatomical surface area. The number of B-lines in every LIS was counted to document potential interstitial lung abnormalities. If they were confluent, the semiquantitative rule suggested by Gargani et al. and Volpicelli et al. was used, that is the percentage of scanning sites occupied by B-lines divided by 10 (i.e., 30% of white screen corresponds to 3 B-lines, 40% to 4 B-lines, and so on) [25,30,32]. When an abdominal organ was seen in the intercostal zone and, therefore, B-lines were absent, it was scored as zero. For the lateral and dorsal zones, the identification of the abdominal organs was used for the identification of the lung, as suggested by the European Respiratory Society. If less than 5 B-lines were observed, it was considered to be negative for the presence of an interstitial syndrome. Marked interstitial changes have previously been defined as the presence of more than 50 B-lines [33]. Additionally, with the help of the application, the topographic prevalence and anatomic distribution of the interstitial abnormalities were recorded, together with the presence of other lung and pleural abnormalities such as an interrupted or thickened pleural line, the presence of subpleural nodules, and the presence of pleural effusions.

### 2.2. Statistical Analysis

Descriptive statistics were performed using R (Version 4.2.2). Normality was assessed using the Shapiro–Wilk test. Non-normally distributed results are reported as medians with interquartile ranges. Normally distributed results are reported as means with standard deviations. Spearman’s rank order correlation was used to assess the correlation between two linear variables. The Wilcoxon test was used for bivariate analysis if the distribution was non-normal. The Kruskal–Wallis test by rank was used for multivariate multiple groups with a non-normal distribution. A significance level of α = 0.05 was used. Correction for multiple testing was performed using the Bonferroni correction, when applicable.

### 2.3. Creation of Figures

All sonographic images were made with the Philips Lumify portable ultrasound system. Figure 1 was created using R (Version 4.2.2). Figure 2 was made using the images collected from the Philips Lumify Platform. Figure 3 was created using Adobe Illustrator 2023.

## 3. Results

### 3.1. Baseline Characteristics and Statistical Analysis

We included 75 consecutive patients with RA presenting to the rheumatology outpatient clinic. The baseline characteristics and demographics are shown in Table 1. Patients were predominantly female (62.3%), seropositive (RF and/or ACPA positive) (82.7%), and ever-smokers (62.7%). Twelve percent were active smokers upon inclusion. The mean age at inclusion was 63.5y (11.7); the mean age at RA diagnosis was 49.1y (15.2). The median RA duration was 12.4 years. All patients received anti-inflammatory therapies, and most were treated with a classical synthetic disease-modifying antirheumatic drug (DMARD) (80%). There were 40% who additionally received a biological DMARD, 5% who additionally received a targeted synthetic DMARD, and 11% who also received corticosteroids.

Older age at inclusion (ρ = 0.36, *p* ≤ 0.01) and older age at RA diagnosis (ρ = 0.42, *p* ≤ 0.01) were both correlated with a higher number of B-lines. DAS-CRP was not correlated with the number of B-lines (*p* = 0.52), nor were the VAS pain or fatigue scores (*p* = 0.98 and *p* = 0.37, respectively). There were no differences in the median number of B-lines based on seropositivity (*p* = 0.93). The median number of b-lines in seronegative patients (RF and ACPA) was 5 (IQR 13) and in seropositive (RF or ACPA) patients 5 (IQR 12.8). No differences in B-lines were seen between male (median 6, IQR 14.2) and female patients (median 4, IQR 12) (*p* = 0.61). No differences in the number of B-lines were observed per treatment group (*p* = 0.61).

The number of pack years did not correlate with the number of B-lines (*p* = 0.87). There was no difference between the median number of B-lines in ever-smokers (median 5, IQR 6.25) when compared to never smokers (median 6, IQR 15.5) in our cohort (*p* = 0.60). Furthermore, we did not detect more pleural abnormalities in smokers than in nonsmokers (*p* = 0.29).

### 3.2. LUS Patterns

We observed several ultrasound patterns such as: normal aeriated lung, interstitial abnormalities as suggested by the presence of a substantial number of B-lines, pleural abnormalities, pleural effusions, and subpleural nodules. The results are summarized in Table 2. In 34 patients (45.3%), pleural abnormalities were observed. Subpleural nodules were observed in 11 patients (14.7%). Thirty-five (46.7%) patients had less than 5 B-lines, fifteen (20%) patients had between 5 and 10 B-lines, eleven (14.6%) between 10 and 20, ten (13.3%) between 20 and 50, one (1.3%) between 50 and 100, and three (4%) of patients had more than 100 B-lines. This translates to the presence of marked interstitial changes in 5.3% when using a cut-off of 50 B-lines. The distribution is shown in Figure 1. In one (1.3%) patient, a voluminous pleural effusion, with subsequent atelectasis of the lower lobe, was noted. The ultrasound findings are shown in Figure 2.

We recorded a complete mapping of the distribution of the B-lines in 50 patients. A heatmap, shown in Figure 3, shows the most-prevalent localizations for B-lines, both in a cumulative manner, where all the B-lines of all patients were added, as well as by stating the number of patients with at least one B-line per intercostal space.

**Figure 1 diagnostics-13-02986-f001:**
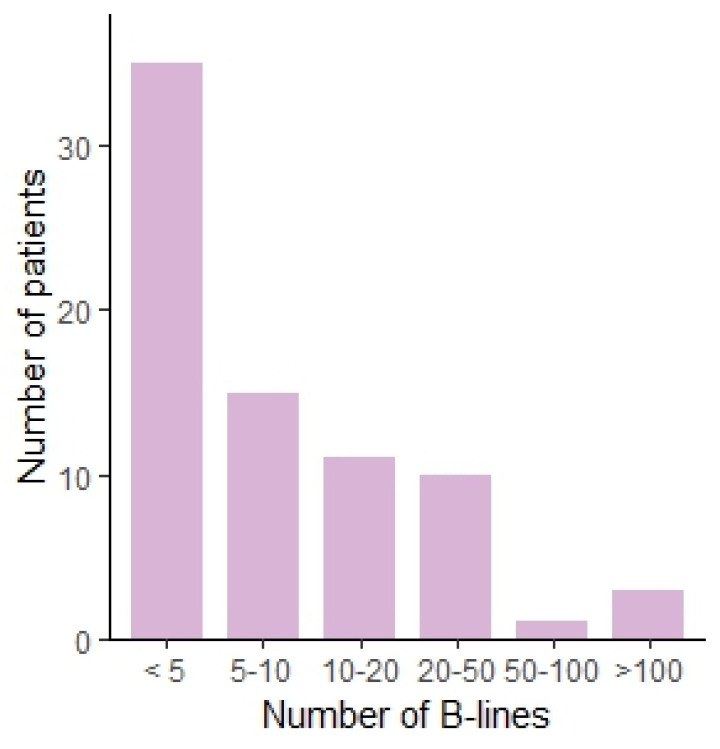
Bar chart showing the number of patients per number of B-lines, categorized.

**Figure 2 diagnostics-13-02986-f002:**
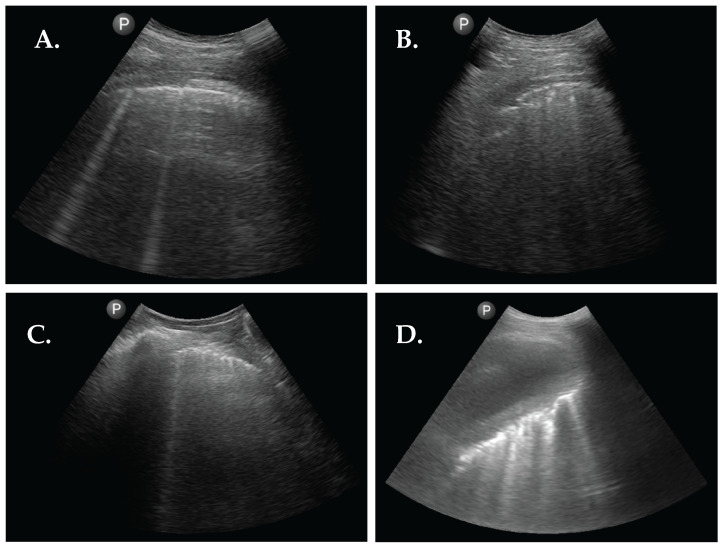
Abnormal sonographic findings from 4 patients with RA. (**A**) LUS image showing 2 B-lines. (**B**) LUS image showing pleural interruptions and a subpleural nodule. (**C**) LUS image showing pleural line thickening and 1 B-line. (**D**) LUS image showing a large pleural effusion with atelectasis of the right lower lobe with multiple inlying B-lines.

**Figure 3 diagnostics-13-02986-f003:**
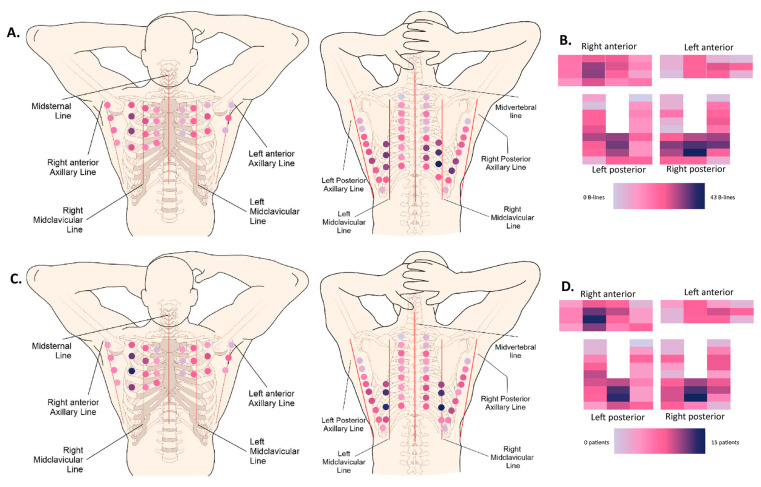
Heatmaps. (**A**) Anatomical (72-zone) heatmap based on the cumulative absolute number of B-lines (50 patients). (**B**) Schematic (72-zone) heatmap based on the cumulative absolute number of B-lines (50 patients). Every rectangle corresponds to one intercostal space localization. (**C**) Anatomical (72-zone) heatmap based on the number of patients that scored positive (>1 B-line) per intercostal space (50 patients). (**D**) Schematic (72-zone) heatmap based on the number of patients that scored positive (>1 B-line) per intercostal space (50 patients). Every rectangle corresponds to one intercostal space localization.

The completion of the thoracic ultrasound examination, examining all 72 intercostal spaces, took between 6 and 12 min, approximately.

## 4. Discussion

In 75 patients with RA, LUS was performed. LUS is a noninvasive and radiation-free tool that can be used bedside or in an outpatient clinic setting. This study showed the array of sonographic findings in RA patients. We detected marked interstitial syndromes as defined by the presence of more than 50 B-lines (5.3%), pleural abnormalities (45.3%), and subpleural nodules (14.7%) in patients with RA presenting to the rheumatology outpatient clinic. Pleural abnormalities consisted of interrupted, irregular, or thickened pleural lines [33]. One patient presented with a voluminous pleural effusion, with inlying fibrinous material. More B-lines were observed in older patients and patients who received the diagnosis of RA at an older age. We saw no association between the number of B-lines and the DAS-CRP score, their smoking history, whether they were seropositive (RF or ACPA) or not, and their current treatment strategy. We have to note, however, that the numbers of patients in the individual treatment group were rather small. There was no difference in the incidence of pleural abnormalities in smokers when compared to nonsmokers.

In patients with systemic sclerosis [20,21,22,23,24,30], where LUS has been partially validated, two cut-offs have been proposed to be indicative of an interstitial syndrome: more than 5 B-lines or more than 10 B-lines [34]. In our study, 46,7% of patients had less than 5 B-lines and 66.7% of patients had less than 10 B-lines. Marked interstitial changes have previously been defined by multiple studies as having more than 50 B-lines [34]. In our cross-sectional cohort, 5.3% complied with this definition. Visually, hotspots for the presence of B-lines are situated bilaterally in the posterior subscapular regions (LISs 7, 8), as well as the anterior right mid-clavicular region (LISs 3, 4, 5). These hotspots were present in both the cumulative heatmap, where all the B-lines of all patients were added, as well as the heatmap showing the number of patients with at least one B-line per intercostal space. Vasquez et al. previously described an association between the depiction of B-lines in ultrasound and the presence of ILD in one of our cohort’s most-frequently affected intercostal spaces, namely the eighth right subscapular space (OR 16.5) [35].

This is the first paper using the 72-window approach, which provided a mapping of findings suggestive of an interstitial syndrome per intercostal space. Secondly, a limitation of the use of ultrasound is the operator dependence. All ultrasounds were performed by the same sonographer, who received training from the European Respiratory Society. Kumar et al. examined the presence of an interstitial syndrome represented by B-lines in SARS-CoV-2 patients and showed substantial interrater reliability for normal thoracic ultrasound (absence of B-lines, absence of pleural abnormalities, absence of subpleural nodules, absence of consolidations) (Cohen κ = 0.79) and the presence of B-lines on LUS (Cohen κ = 0.79) [36]. Interrater reliability was lower for the presence of pleural abnormalities and subpleural nodules. In systemic-sclerosis-associated interstitial lung disease, Gutierrez et al. reported similar interrater reliabilities for simplified LUS assessments of B-lines (Cohen κ = 0.769–0.885) [37]. On the other hand, the reported agreement concerning pleural abnormalities was lower (Cohen κ = 0.54) [34]. This warrants our results concerning the pleura to be interpreted with more caution, especially since a clear consensus definition of pleural abnormalities is currently lacking. Furthermore, we used a curvilinear probe to examine the pleural line, which further supports that caution is needed for the interpretation of the pleural abnormalities as linear probes have a higher resolution when examining the pleura. Nevertheless, the evaluation of the pleural line could play a possible role in LUS screening protocols, as supported by Kumar et al. [38]. Future studies could combine the use of linear probes and curvilinear- probes, with linear probes providing a higher resolution for more-superficial tissues. Another limitation of this study is the limited number of patients (n = 75) and its cross-sectional setup. Even so, we included patients with a broad range of RA duration, which is an advantage given the generalizability of our findings. Future studies should focus on larger sample sizes and prospective setups. These studies could contribute to the understanding of a possible temporal link. An additional limitation is the absence of a comparison with high-resolution CT scans and pulmonary function testing, to further explore the significance of our ultrasound findings. This was, however, out of the scope for this article as the aim was limited to the description of our sonographic findings and to prove the feasibility of thoracic ultrasound in an outpatient rheumatology clinic. The feasibility of LUS in this setting is reflected by the exam duration of a maximum of 12 min.

The main strength of this paper is that it provides an extensive basis for future trials examining the role of LUS in the detection of extrapulmonary manifestations of RA and, more specifically, RA-ILD. As the presence of RA-ILD is associated with high mortality and morbidity and is currently only detected at a late stage, the presence of an interstitial syndrome on ultrasound could be an important piece of the puzzle for future screening trials. Furthermore, LUS can play a role in the detection of pleural effusions and in defining the presence of small subpleural nodules. Future studies focusing on the comparison of LUS to HRCT and pulmonary function testing are highly needed. These future studies should also look into the development of simplified screening protocols by focusing on intercostals spaces, which were more frequently affected in our visual heatmaps.

## 5. Conclusions

This study explored different thoracic ultrasound findings in RA patients. In this cohort of 75 patients, approximately half of all RA patients had some degree of pulmonary abnormalities on thoracic ultrasound. There were 53.3% of the RA patients who had an interstitial syndrome (>5 B-lines). Visually, hotspots for the presence of B-lines were situated bilaterally in the posterior subscapular regions, as well as the anterior right mid-clavicular region. Moreover, pleural effusions (1.3%), pleural abnormalities (45.3%), and pleural nodules (14.7%) were observed in our RA patient population. Additionally, LUS has proven to be a feasible tool in the outpatient rheumatology clinic. Further research is needed to validate the diagnostic accuracy of lung ultrasound in patients with rheumatoid arthritis, when compared to high-resolution CT scans and pulmonary function tests.

## Figures and Tables

**Table 1 diagnostics-13-02986-t001:** Baseline characteristics of all 75 patients with RA having undergone thoracic ultrasound.

	*n* (%)
**Female**	47 (62.67%)
**Seropositive**	62 (82.67%)
**RF-positive**	45 (60%)
**ACPA-positive**	57 (76%)
**Ever-smoker**	47 (62.67%)
**Active smoker**	9 (12%)
**Current treatment**	
csDMARD	24 (32%)
csDMARD + bDMARD	30 (40%)
bDMARD	8 (11%)
csDMARD + tsDMARD	4 (5%)
CS + csDMARD	8 (11%)
CS	1 (1.33%)
	**Median—IQR**
**Age at inclusion**	64.17 (16.79)
**Age at RA diagnosis**	46.75 (22.12)
**RA duration**	12.35 (14.40)
**DAS-CRP**	2,2 (1.25)
**VAS pain**	40 (50)
**VAS fatigue**	35 (60)
**Pack years if smoker**	15 (22)

RF = rheumatoid factor, ACPA = anti-citrullinated protein antibody, csDMARD = conventional synthetic disease-modifying antirheumatic drug, bDMARD = biological disease-modifying antirheumatic drug, tsDMARD = targeted synthetic disease-modifying antirheumatic drug, CS = corticosteroids, RA = rheumatoid arthritis, DAS-CRP = disease activity score-C-reactive protein, VAS = visual analogue scale.

**Table 2 diagnostics-13-02986-t002:** Overview of lung ultrasound findings.

	*n* (%)
B-lines	
<5	35 (46.7%)
5–10	15 (20%)
10–20	11 (14.6%)
20–50	10 (13.3%)
50–100	1 (1.3%)
>100	3 (4%)
Pleural abnormalities	34 (45.3%)
Subpleural nodules	11 (14.7%)
Pleural effusion	1 (1.3%)

## Data Availability

Data generated or analyzed during this study are available from the corresponding author upon reasonable request.

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
