# Peer review of "Ultrasonographic Presentation and Anatomic Distribution of Lung Involvement in Patients with Rheumatoid Arthritis"

_diagnostics, 2023, doi:10.3390/diagnostics13182986_

Round 1

Reviewer 1 Report

Comments and Suggestions for Authors

I read with interest the manuscript by Vermant et al.. The topic is fascinating to perform an early diagnosis of lung involvement in RA. Hereby I report my comments/questions. 

- Are pleural abnormalities associated with demographic and/or disease activity scores? I could not find any statistical analysis. Is there any associations among clinical features and ultrasound lesions? 

- May it be possible that smokers may present more pleural abnormalities or B line compared to the non-smokers? 

- No healthy controls have been included in this study. It would be interesting to assess wheather lung/pleural abnormalities are detectable both in general population and in RA patients. 

- Do you perform also CT scans and spirometry in a subset of these patients? Would it be interesting to assess the role of ultrasound compared to other methods ( X-rays, CT scan, spirometry, O2 saturation, histopathological analysis). 

Reviewer 2 Report

Comments and Suggestions for Authors

Vermant M et al described the ultrasonographic abnormality and distribution in patients with RA.They found B-lines presence and pleural irregularity are major features, indicated interstitial lesion and plerual involvement. However, there are some limitations in this article. 

1. The lack of Chest HRCT scanning (gold standard of imaging for pulmonary disease) and/or pulmonary function test;

2. How to confirm the sonographic features are related to RA, or infection or drug toxicity;  

3. Only one observer assessed and analysed the sonographic finding;

4. A total of 72 LIS were examined for comprehensive US assessment. Average time-consuming for every patient?  The effect of abdominal organs on the examination?

Round 2

Reviewer 2 Report

Comments and Suggestions for Authors

I agree with the author's response and accept the current modified version.